# Climate Change Adaptation Strategies at a Local Scale: The Portuguese Case Study

**DOI:** 10.3390/ijerph192416687

**Published:** 2022-12-12

**Authors:** Margarida Ramalho, José Carlos Ferreira, Catarina Jóia Santos

**Affiliations:** 1Department of Environmental Sciences and Engineering, NOVA School of Science and Technology, NOVA University Lisbon, Campus da Caparica, 2829-516 Caparica, Portugal; 2ARNET—Aquatic Research Network/MARE—Marine and Environmental Sciences Centre, Campus da Caparica, 2829-516 Caparica, Portugal

**Keywords:** coastal areas, community-based adaptation, nature-based solutions, green infrastructure

## Abstract

Coastal areas are home to more than 2 billion people around the globe and, as such, are especially vulnerable to climate change consequences. Climate change adaptation has proven to be more effective on a local scale, contributing to a bottom-up approach to the problems related to the changing climate. Portugal has approximately 2000 km of coastline, with 75% of the population living along the coast. Therefore, this research had the main objective of understanding adaptation processes at a local scale, using Portuguese coastal municipalities as a case study. To achieve this goal, document analysis and a questionnaire to coastal municipalities were applied, and the existence of measures rooted in nature-based solutions, green infrastructures, and community-based adaptation was adopted as a variable. The main conclusion from this research is that 87% of the municipalities that answered the questionnaire have climate change adaptation strategies implemented or in development. Moreover, it was possible to conclude that 90% of the municipalities are familiar with the concept of nature-based solutions and all the municipalities with adaptation strategies include green infrastructure. However, it was also possible to infer that community-based adaptation is a concept that most municipalities do not know about or undervalue.

## 1. Introduction

In the last few decades, climate change has become a major topic of discussion all around the world. Climate change will have long-term effects on human lives and beings, due to the drastic changes in the ecosystem’s patterns and processes. In some cases, this has already led to changes in some economic activities [1,2,3].

To address climate change and the consequences that derive from it, various authors have produced the concept of climate change adaptation, varying in their approach to the concept, which has resulted in various definitions [4,5,6,7,8]. Schmidt-Thomé (2017) [4] explores the concept of climate change adaptation, mentioning that the concept reflects the context in which it is applied and giving the example of the definition by the United Framework on Climate Change: “practical steps to protect countries and communities from the likely disruption and damage that will result from effects of climate change”.

Moreover, Smit and Pilifosova (2001) [6] define adaptation as “changes in processes, practices, or structures to moderate or offset potential damages or to take advantage of opportunities associated with changes in climate”.

However, the more consensual definition is the one by the Intergovernmental Panel on Climate Change [8], which specifies that climate change adaptation can be defined as “the process of adjustment to actual or expected climate and its effects in order to moderate harm or take advantage of beneficial opportunities (…)”.

Since the effects of climate change and global warming have become more severe in the last decade, many countries have decided to implement national and local strategies for climate change adaptation, to better prepare themselves for the adverse consequences that can be projected for each area [9,10]. It has also been found that climate change impacts are mainly experienced at a local level, which helped to prompt the mentioned strategies [11].

Climate change adaptation has more success when applied at a local scale, contributing to bottom-up approaches instead of top-down approaches, and the potential transferability of the best adaptation processes [12]. As such, climate change policies, including adaptation strategies, must be local, mainly because of the different contexts provided by a variety of community stakeholders, which contribute to the proximity to the challenges and an understanding of bigger problems at a local scale [11,13]. The implementation of climate change adaptation at a local level is illustrated in studies undertaken by Roberts (2008), Laukkonen et al. (2009), and Rauken et al. (2014) [14,15,16].

It is important to note that larger cities in developed countries have easier access to resources for climate change adaptation planning—benefiting from the presence of research institutions, conferences, and other entities. Moreover, smaller cities generally have fewer resources and have more difficulty being engaged in national and international networks [17]. Therefore, it is crucial to evaluate case studies of implemented climate change adaptation strategies to identify current good practices and, simultaneously, overcome discrepancies such as the ones mentioned.

Climate change adaptation options can include measures derived from the concepts of nature-based solutions (NBS), green infrastructure (GI), or community-based adaptation (CBA). Some uses of NBS include green building–integrated systems and technologies—green roofs and walls [18,19], green parks [20], permeable pavements, and stormwater ponds [21,22].

According to several authors [23,24,25], GI can have several benefits that contribute to the adaptation and mitigation of climate change, such as air quality improvement, carbon storage increase, urban noise reduction, and stormwater management.

Jarillo & Barnett (2021) and Regmi et al. (2015) mention that CBA tackles adaptation by learning from the experiences of local communities with climate change effects, identifying the problems, and coming up with activities that capacitate the people [26,27].

This research aims to draw conclusions on adaptation processes at a local scale, using the Portuguese study case, and assess how to replicate it in a global context, since it is understood that local adaptation planning is the way most communities can adapt to climate change effects.

Additionally, this research also intends to make a national survey of the local adaptation strategies of coastal municipalities in Portugal, and their inclusion of nature-based solutions, green infrastructure, and community-based adaptation. This is particularly relevant considering the new Portuguese Climate Law (Lei de Bases do Clima) [28], which mandates municipalities to have local adaptation strategies by the end of 2023.

## 2. Research Framework

### 2.1. Climate Change Adaptation in the European Union

The first European Union (EU) adaptation strategy was presented in April 2013. This document had three main objectives: promoting action by the Member States, promoting better-informed decision-making, and promoting adaptation in key vulnerable sectors [16].

The implementation of the strategy was based on eight actions [29]:Encourage all Member States to adopt comprehensive adaptation strategies by providing guidelines and identifying key indicators to measure Member States’ level of readiness;Provide LIFE funding to support capacity building and step-up adaptation action in Europe (2014–2020);Introduce adaptation in the Covenant of Mayors framework (2013/2014), especially to support adaptation in cities;Bridge the knowledge gap by working with the Member States and stakeholders to identify tools and methodologies that can address adaptation knowledge gaps;Further develop Climate-ADAPT as the “one-stop shop” for adaptation information in Europe through the improvement of access to information between this portal and other relevant platforms;Facilitate the climate-proofing of the Common Agricultural Policy (CAP), the Cohesion Policy, and the Common Fisheries Policy (CFP) by integrating climate change adaptation measures in these documents, and by capacitating relevant stakeholders in this process;Ensuring more resilient infrastructure by mandating European standardization organizations to map industry-relevant standards in the areas of energy, transport, and buildings and to identify standards that can better include adaptation considerations. The Commission also provided guidelines to climate-proof vulnerable investments and additional guidance for its Communication on Green Infrastructure;Promote insurance and other financial products for resilient investment and business decisions.

In 2018, the 2013 Strategy was evaluated to understand the progress the Member States had made since its implementation. The main conclusions from the evaluation drew on the need for a greater focus on adaptation efforts at the EU level, specifically on water and drought, climate change adaptation at a local and urban level, agriculture policy, climate finance, insurance, and business. It is also concluded that the EU Strategy has likely enhanced the political focus of Member States on adaptation issues, in addition to the Paris Agreement [30].

In 2021, the European Commission released a new EU Strategy for Adaptation to Climate Change. This document has three main objectives, building on the 2013 strategy and its evaluation [31]:Make adaptation smarter by improving knowledge and availability of data;Make adaptation more systemic through the support of policy development;Speed up adaptation across the board by accelerating the development and rollout of adaptation solutions.

With these goals in mind, the strategy presents the long-term vision for the EU in terms of adaptative capacity to minimize vulnerability to the effects of climate change while being in synergy with other Green Deal policies. It is also noted by the Commission that there is an urgent need to develop effective and inclusive governance mechanisms that can connect policymakers and scientists [31].

Through this document, the importance of the Climate-ADAPT Platform is also reinforced, which the EU aims to make the authoritative European platform for climate change adaptation. Climate-ADAPT is a partnership between the European Commission and the European Environment Agency (EEA), which aims to share data and information about adaptation, the national adaptation strategies and actions of the Member States, case studies, and tools that support adaptation planning [32].

### 2.2. Adaptation in Portugal

Portugal has had a National Strategy for Climate Change Adaptation (ENAAC) since 2010, introduced by resolution no. 24/2010 of the Council of Ministers of 1 April 2010 [33]. The main objectives of this strategy were the following:Information and knowledge by developing a scientific a technical basis of information;Reduce vulnerability and increase the capacity to respond by identifying and defining priorities in terms of climate change adaptation measures;Participate, raise awareness, and publicize through the contribution of stakeholders;International cooperation by approaching the national responsibilities in terms of cooperating with international adaptation policies.

In 2015, the same strategy was renewed, originating ENAAC 2020, which was supposed to run until 2020 but was extended to 2025 due to the approval of the National Plan of Energy and Climate (PNEC 2030). According to resolution no. 53/2020 of the Council of Ministers of 10 July 2020 [34], this strategy is also accompanied by an action plan, which defines concrete lines of action for adaptation to climate change.

ENAAC 2020 was approved in resolution no. 56/2015 of the Council of Ministers of 30 July 2015 [35], renewing ENAAC 2010–2013 and with the following main objectives:Better the knowledge level about climate change by updating and developing information about climate change, while assessing its risks, impacts, and consequences. This material must be exposed through communication platforms and awareness campaigns;Implement adaptation measures through two options: consulting stakeholders from the sectorial working groups and through the collection of information regarding best practices, both on a national and international level (especially south European countries);Promote the integration of adaptation in sectorial policies.

The strategy includes six thematic areas, which were selected through the knowledge obtained in the ENAAC 2010–2013, which are seen in most activity sectors. The areas are research and development (R&D), financing, international cooperation, communication, territory planning, and water resources management.

Moreover, the nine updated priority sectors are the basis of ENAAC 2020 and are as follows: agriculture, biodiversity, economics, energy, forests, health, security of people and goods, transport and communications, and coastal areas.

For the mentioned sectors, and as it happened in the first ENAAC, a working group was defined by the competent public authority.

According to this strategy, most scenarios for 2080–2100 for Portugal project the following climate change effects:A general increase in the average yearly temperature in every region of the country;An increase of up to 3 °C for the highest temperature in summer, for the coastal areas, and an increase of up to 7 °C for the countryside. For the Madeira and Azores islands an increase between 1 °C and 3 °C is projected;A reduction in frost days and an increase in hotter days and tropical nights;An increase in forest fire risk, change of land use, and implications for water resources;A significant change in the precipitation cycle, which may include a reduction in precipitation during spring, summer, and autumn in mainland Portugal. There is also a possibility of a decrease in yearly precipitation and an increase in winter rainfall, due to the rise in the number of days with stronger rain.

Due to the mentioned projections, as well as other factors, the Portuguese Government presented the new Portuguese Climate Law of 31 December 2021 [28]. This law establishes several objectives related to the environment and the climate emergency, among them the need to “reinforce resilience and the national capacity to adapt to climate change”.

Climate change causes more relevant impacts at a local scale and as a result, there has been a continuous effort to shift policies from national and regional levels to the local one [36]. In view of the new Climate Law [28], each Portuguese municipality must have adaptation strategies and action plans implemented by the end of 2023.

As mentioned earlier in the chapter, for the present paper, ClimAdaPT.Local assumes relevance in the present paper. This program was created in 2016, and its main goal is to encourage local climate change adaptation in Portugal, through the following steps [32]:Facilitate experience exchange between municipalities;Promote knowledge exchange between municipalities, universities, and research centers, as well as local companies;Promote international cooperation relations;Promote the empowerment of municipalities.

### 2.3. Natured-Based Solutions

Due to the more frequent effects and consequences of extreme phenomena caused by climate change, there has been a focus on implementing local-scale climate change adaptation actions that help reduce the vulnerability of the receiving environment [36,37].

These interventions make up the concept of nature-based solutions (NBS), which can be defined, broadly, as solutions that are inspired and supported by nature and are cost-effective, while simultaneously providing environmental, social, and economic benefits and building resilience while bringing more nature and diverse natural features and processes into urban spaces through resource-efficient and systemic interventions [21,38,39,40].

Moreover, NBS are interventions that are supported by nature, which can evolve and change. This implies the active management of these systems to ensure that their services are provided [41].

### 2.4. Green Infrastructure

Green infrastructure (GI) is a relatively recent concept that originated in the 1990s, associated with green spaces [42]. One of its different definitions is that it is an interconnected network of green space that conserves natural ecosystem values and functions and provides associated benefits to human beings. However, GI has both an ecological and engineering approach to it and is becoming a priority for decision-makers [42,43].

Additionally, GI can aid in adaptation to climate change in three main aspects: urban heat island effect—by regulating temperature in urban spaces, especially in population-dense locations [44,45,46]; flood risk management—through green cover that can reduce water runoff [44,47]; and ecosystem resilience—by preventing ecosystem fragmentation while increasing the number of protected areas and maintaining habitat connectivity [44,48].

Furthermore, the more widely accepted definition of GI comes from the European Commission (2013) and can be defined as “a strategically planned network of natural and semi-natural areas with other environmental features designed and managed to deliver a wide range of ecosystem services [29]”.

On land, GI is present in rural and urban settings, incorporating green spaces (or blue, if aquatic ecosystems are concerned) and other physical features in terrestrial areas [29]. GI can have an ecological approach, for instance in the form of a natural system composed of national parks, parkways, forests, community gardens, and green corridors, among others [49]. Because of this, GI can respond to a wide range of environmental, social, and economic challenges, including climate change adaptation [42].

NBS and GI are closely related concepts, being complementary to a certain degree. NBS entails a more holistic perspective, aiming to support the implementation of solutions that approach biodiversity conservation, ecosystem service protection, and green infrastructure [50,51]. As such, GI is an important component of NBS.

### 2.5. Community-Based Adaptation

The community-based adaptation (CBA) concept has evolved in recent years, due to the more frequent impacts of climate change, especially in coastal communities, namely the rise in the sea level. These accumulated effects and consequences have become the drive needed for the development of community-based approaches in terms of climate change adaptation [52,53].

CBA happens in local communities vulnerable to the impacts of climate change by identifying, assisting, and implementing activities that will help strengthen the adaptative capacity of these populations [54]. CBA is a bottom-up approach to climate change adaptation with the aim of enhancing adaptive capacity to climate change [55]. As such, the activities mentioned are usually related to participatory processes, connecting local stakeholders and the local communities in the reduction of the risk associated with the rising effects of climate change [56,57].

Since CBA is also a place-based approach, the planning process for the communities must consider the social and ecological dynamics and priorities. According to Basel et al. [58], the integration of the CBA process “is achieved through a high level of trust and community engagement and input, over a long duration to establish meaningful relationships and understand community priorities and drivers”.

Nevertheless, this process is not without criticism. One recurrent criticism of CBA is that there is a need to make the process more relevant to risks and policies outside of communities, that is, on a bigger scale—upscaling [54]. Another criticism is that CBA’s techniques can be difficult to replicate, due to their adaptation to the specifics of the community they are applied to. Additionally, there are concerns about how to make this process a mainstream approach, in a way that makes decision-makers adapt it through policies [54].

Forsyth (2017) also draws on another CBA challenge: its capacity for representing local people fairly, and the simplistic way the term “community” can be treated. Community implies people in a certain locale act as a unit, which rarely happens—with communities having internal divisions. This can also contribute to the difficulties in implementing CBA, namely in its participatory component [59].

## 3. Material and Methods

### 3.1. Methodology

The methodology presented in this section has four main steps, as described in Figure 1. Each of these steps will be explained in depth later in this chapter.

#### 3.1.1. Step 1—Definition of the Study Area

The study area defined for the purpose of this research is the coastal municipalities in Portugal, which also include the coastal municipalities of the islands of Madeira and the Azores, as seen in Figure 2.

The coastline of Portugal, including the Azores and Madeira islands, is approximately 2000 km long and, as such, 75% of the Portuguese population is concentrated on the coast. This area also generates roughly 80% of the Portuguese Gross Domestic Product (GDP), which proves its importance on a national level [60].

Due to the length of its coast, as well as its exposure to the sea waves from the North Atlantic, Portugal is one of the countries most affected by coastal erosion in Europe. With the increasing frequency of events caused by climate change, it is expected that coastal erosion will worsen on the Portuguese coast [61].

Therefore, it is necessary to establish and implement measures and solutions to adapt to coastal cities and towns of the country. Such measures and solutions are one of the bases of the current study.

#### 3.1.2. Step 2—Documentary Analysis

One of the first actions related to this research was the making of an inventory of all the Portuguese coastal municipalities and their work towards climate change adaptation so far.

To complete this inventory, it was necessary to read and analyze the adaptation strategies and plans made by each municipality. It is important to note that not all coastal municipalities have implemented climate change adaptation strategies. This will be presented in the following chapter, which is dedicated to the results of this study.

In this step of the methodological framework, the authors of this research aimed to answer the following questions to categorize the municipalities’ adaptation strategies and plans:Is the municipality a participant in the ClimAdaPT program?What was the methodology used for the elaboration of the strategy?What are the main climate change projections for the municipality?Does the adaptation strategy mention green infrastructure or infrastructure measures?How many adaptation options are mentioned in the strategy?What is the priority of green infrastructure among the other adaptation options?Does the adaptation strategy plan how to integrate the strategy into the territory management instruments (IGT)?

It is important to note that questions 5 and 6 were only asked if the municipality was a part of the ClimAdaPT program. This happened because all ClimAdaPT’s participants are required to follow a similar structure for their adaptation strategy structure.

#### 3.1.3. Step 3—Inquiries to the Coastal Municipalities

To achieve the purpose of this research, there was a need to undertake a survey of the adaptation strategies of coastal municipalities in Portugal.

The initial survey was directed to all 92 municipalities on the Portuguese coast, including Madeira and the Azores islands. For these municipalities, the questions asked focused on the adaptation strategies and their respective inclusion of green infrastructure, nature-based solutions, and community-based adaptation.

The inquiries were made on Google Forms, since this tool also allows the export of data and, consequently, better data treatment.

The inquiries were sent to all 92 municipalities via the author’s institutional e-mail, with the period of the responses being between 3 June and 7 October.

The inquiries were divided into the following sections:General information about the strategy of the municipality;Green infrastructure and nature-based solutions and their inclusion in the strategy;Community-based adaptation;

The last two sections of the inquiry were only presented if the municipality answered that they had implemented a climate change adaptation strategy.

#### 3.1.4. Step 4—Data Treatment

After obtaining the responses to the inquiry, the data were explored further using Microsoft Excel.

As such, a group of parameters was explored, namely:General data concerning the inquiry;Data from municipalities that answered they do not have an adaptation strategy;Data from municipalities that have an adaptation strategy in development.Data from municipalities that answered they have an adaptation strategy.

For the first point in the list above, the results obtained through the data from the questionnaire were the characterization of the municipalities, in terms of their answers to the inquiries and if said municipalities have a climate change adaptation strategy.

As for the second point, the data acquired allowed us to explore the reasons why municipalities do not have climate change adaptation strategies implemented.

Moreover, the third point reflects on the municipalities that have a strategy in development, and at which stage of development it is.

Lastly, the final point allowed for wider data treatment and more conclusive results, due to extensive questions asked to municipalities with adaptation strategies. As such, it was possible to divide this point into four sub-sections of results: general data, nature-based solutions, green infrastructure, and community-based adaptation.

## 4. Results

### 4.1. Results from the Documentary Analysis

From the documentary research, it was possible to observe that out of all 92 municipalities, only 18 have an individual climate change adaptation strategy available for consultation online. In the following table, the mentioned municipalities are presented, along with their Nomenclature of Territorial Units for Statistical Purposes (NUTS II).

Table 1 shows that Norte is the region with the most individual adaptation strategies, with a total of five, which corresponds to 56% of the coastal municipalities in this NUTS II.

From the municipalities that do not have a climate change adaptation strategy available for consultation online, it was possible to understand that the regions with the fewest strategies are the Azores and Madeira islands, with 95% and 91%, respectively, of their municipalities, not having the documents.

Another parameter observed was the participation of municipalities in the ClimAdaPT project. Approximately 60% of municipalities are participants in this project, meaning that 11 municipalities have elaborated their strategy in compliance with this program and its methodology.

Additionally, the documentary research showed that 44 municipalities are included in either inter-municipal or metropolitan plans, such as the Lisbon and Porto Metropolitan Areas adaptation strategies, the plans from the Madeira and Azores islands, the Algarve and the Oeste inter-municipal adaptation plans, and the inter-municipal plan of the region CIM of Coimbra.

However, if only individual municipal strategies are considered, then the total of municipalities that do not have an adaptation strategy document is 74. This means that 80.4% of Portuguese municipalities do not have an individual adaptation strategy in place.

Another factor that was examined in the climate change adaptation strategies was their inclusion of green infrastructures. It was possible to verify that, of the 18 individual strategies, 15 mentioned and included adaptation options regarding green infrastructure. However, three municipalities did not consider GI in their documents. The three municipalities that did not include GI are all from the Norte: Espinho, Vila do Conde, and Esposende.

### 4.2. Results from the Questionnaire

For this inquiry, all 92 coastal municipalities of Portugal were contacted and invited to answer a questionnaire about adaptation strategies.

#### 4.2.1. General Data

Firstly, it is important to acknowledge that 50% of the Portuguese coastal municipalities answered the questionnaire. Therefore, the results of this evaluation are not completely representative of all coastal municipalities in Portugal and only represent 46 municipalities.

Considering the NUTS II, the NUTS II with the highest percentage of responses to the questionnaire was the Norte, with 66.7% of the municipalities surveyed answering. However, the NUTS II with the smallest percentage of responses was Alentejo, with only 16.7% of the surveyed municipalities of this region answering.

Due to the higher number of municipalities contacted, it was expected that the Metropolitan Area of Lisbon would have a higher percentage response. Nevertheless, this NUTS II had 56.3% of responses out of the 16 municipalities contacted.

Regarding the municipalities from the islands, Azores had responses from 52.6% of its municipalities, while Madeira had 27.3% of responses.

Concerning only the municipalities that answered the inquiry, and if they have a climate change adaptation strategy implemented or in development, the results are described in Figure 3.

From this graphic, it is possible to understand that 13.0% of the municipalities do not have an adaptation strategy while 43.5% do, and the remaining 43.5% have adaptation strategies in development.

Considering the existence of the climate change adaptation strategies, the NUTS II with the fewest strategies is Madeira—with one of the three municipalities not having a strategy implemented.

Regarding strategies in development, it is possible to verify that the NUTS II with the most climate change adaptation strategies in development is the Azores. Out of the 11 answers, 7 municipalities are developing strategies.

As for the region with the most adaptation strategies, the Norte, all the municipalities that answered the questionnaire have strategies in place. After the Norte, the Lisbon Metropolitan Area is the NUTS II with the most municipal adaptation strategies implemented, as six out of the nine municipalities that have answered the inquiry have these documents put into effect.

#### 4.2.2. Municipalities That Do Not Have a Climate Change Adaptation Strategy

Out of the 46 municipalities that answered the inquiry, six of them do not have a climate change adaptation strategy. In the table below Table 2, the municipalities that do not have individual strategies and their corresponding NUTS II are presented.

Olhão Municipality, although part of the inter-municipal plan for climate change adaptation of the Algarve Metropolitan Area (PIAAC AMAL), stated they did not have a climate change adaptation strategy. The specific reason Olhão indicated for not having an individual adaptation strategy is that it did not have an environment/sustainability department until recently.

The same happened with Câmara de Lobos Municipality and Horta and Santa Cruz, which are included in the regional climate change adaptation plans for Madeira and Azores, respectively. These municipalities claimed the following reasons for not having individual adaptation strategies:Lack of decision by the executive and lack of human resources (Câmara de Lobos);The adaptation strategy is not planned (Santa Cruz da Graciosa);Lack of interest by the executive (Horta).Regarding Marinha Grande and Vagos, when asked about the lack of development of the adaptation strategies, the following reasons were given:Lack of human resources;Lack of funding for the elaboration of the document.

#### 4.2.3. Municipalities That Have a Climate Change Adaptation Strategy in Development

To better explore the results obtained through the inquiry, municipalities that have strategies in development were also accounted for. In total, 20 municipalities have climate change adaptation strategies in development, which are presented in the following table alongside the corresponding NUTS II.

Table 3 shows that, apart from the municipalities from the Norte region, all of the NUTS II regions have municipalities with strategies in development. The Norte region is not represented because all the municipalities that responded to the inquiry, and that are from this NUTS II, have strategies implemented.

To study how far along the climate change adaptation strategies are, results showing their stages of development are presented in Figure 4.

From the graphic in Figure 4, it is possible to observe that 10 municipalities have their strategy documents in elaboration, while 7 municipalities have the strategy in preparation, and 2 municipalities admit that the process is delayed. In this last parameter, it was not possible to verify why the process is delayed.

Moreover, these results show a clear interest from these municipalities in developing an adaptation strategy to comply with the New Climate Law and to implement measures that can adapt these municipalities and avoid a higher risk.

#### 4.2.4. Municipalities That Have a Climate Change Adaptation Strategy

As already mentioned, 20 municipalities out of the 46 municipalities that answered the inquiry have a climate change adaptation strategy implemented. These municipalities are presented in Table 4, as well as their corresponding NUTS II.

From Table 4, it is possible to observe that, according to the answers to the inquiry, no municipality from Alentejo and Madeira islands has a climate change adaptation strategy.

Regarding the remaining answers, the results presented show that they are well distributed within the NUTS II, with representation in five out of the seven NUTS II regions.

Through the questionnaire, it was possible to obtain the dates when the strategies of the municipalities above were published/elaborated, with the results being shown in Figure 5.

From this graphic, it is shown that 2016 was the year when most adaptation strategies were launched, followed by 2017, 2019, and 2020.

The responses to the inquiries also allowed us to ascertain the scale of the strategy—whether it acts at a municipal or inter-municipal scale. Figure 6 displays the results.

Through Figure 6, it is possible to verify that most adaptation strategies act on a municipal level. However, three municipalities have strategies that act on an inter-municipal scale, and the scale of the remaining two municipalities is metropolitan.

The responses to the questionnaire also made it possible to understand at what phase the adaptation strategies from the municipalities are. As such, 11 municipalities have concluded their document, while the remaining 7 are in the process of executing their strategies.

As stated earlier in this section, through the inquiry, the municipalities were questioned about their inclusion of nature-based solutions in their climate change adaptation strategies. The following graphic presents the number of municipalities that utilized this measure in their documents.

As presented in Figure 7, 18 municipalities consider nature-based solutions in their strategies, and 2 municipalities do not. As per this distribution in NUTS II, it was verified that the two municipalities that do not include NBS in their strategies are from the Norte region, namely, Vila do Conde and Matosinhos. The remaining 18 municipalities are distributed through all the Portuguese NUTS II, except the Madeira islands and Alentejo.

The motives stated by Vila do Conde and Matosinhos for the lack of NBS in their strategies were the following: even though NBS were not explicit in the document, they are implicitly integrated into the adaptation strategy, and the NBS were already contemplated in another action plan from the municipality.

Regarding specific NBS measures adopted in the climate change adaptation strategies, the results are presented in Figure 8.

Figure 8 shows that 89% of the municipalities that have adaptation strategies expect to recover and restore water lines in their territory, 78% want to restore the ecosystems in their municipality, and 72% hope to accomplish more urban green spaces. As for the other NBS measures mentioned by two municipalities, they encompass sustainable drainage systems.

NBS are well-known to municipalities, as it is possible to see from the above results. In most municipalities, adaptation options consider NBS, whether in its green infrastructure component or in the recovery of ecosystems and water lines.

GI data were also obtained because of the inquiry sent to the coastal municipalities, specifically the ones that answered they have a strategy in motion. All the municipalities that have a climate change adaptation strategy mention and plan to implement GI. As such, Figure 9 presents the results obtained regarding the types of GI the municipalities expect to execute through their adaptation strategies.

Figure 9 shows that the top GI measure for the surveyed municipalities is related to the rehabilitation measures for streams and associated riparian galleries, with 85% of the municipalities having this GI option in their strategy. Moreover, 80% of the municipalities consider options that reinforce green spaces in their territory and specific measures for flood risk management. Most municipalities also regard the promotion of sustainable solutions and initiatives in their strategies and half of the municipalities have innovative sustainability strategies for urban spaces. Lastly, 10% of the municipalities also refer to other GI options, such as the promotion of sustainable agriculture practices and the adaptation of more resilient species in the management of green infrastructure in the territory.

Through the questionnaire, it was possible to verify that 15 municipalities do not include community-based adaptation in their strategies, while 5 do. These five municipalities are Ribeira Grande, Ílhavo, Matosinhos, Loures, and Cascais from the Azores, Centro, Norte, and Lisbon Metropolitan area NUTS II, respectively.

In order to know in which way the municipalities are utilizing the CBA concept, a question was asked to the municipalities about the measures used that can be encompassed by CBA. The results are presented in Figure 10.

Out of all the municipalities that have a strategy, 10% have implemented CBA in their strategies through adaptation options proposed by the local community, 5% have applied projects suggested by the local population, and 20% have projects that require active participation by the community.

Concerning the municipalities that do not include CBA in their strategies, the main reason why municipalities did not include CBA in their adaptation strategies is that they did not understand the need for it. Two municipalities also answered they were unaware of the utility of this concept and, finally, six municipalities referred to other motives. Three municipalities did not answer this question.

The motives included in “others” were the following:The creation of a stakeholders commission already covers the active participation of the local community in strategies such as the climate change adaptation strategy;The municipality followed the ClimAdaPT methodology, which did not include CBA at that date;The municipality is contemplating the creation of a climate change adaptation local network.

## 5. Discussion

### 5.1. Documentary Analysis

Through the results presented in the previous section of the documentary analysis, it was possible to verify that the number of adaptation strategies available for consultation is still small, compared to the number of coastal municipalities in Portugal. Most of the municipalities are included in regional or inter-municipal strategies, which have a tendency to treat the municipalities as a unit. This can result in a lack of detail for the municipalities, on an individual level, or can lead to some municipalities being overlooked in favor of others.

Approximately 60% of municipalities that have climate change adaptation strategies are participants in the ClimAdaPT project. This means that this project was an important factor in the elaboration of the strategies.

To get a better geographical distribution of the participating municipalities, ClimAdaPT invited several municipalities throughout the Portuguese territory, including those from NUTS II that usually do not have the resources (either human resources or financial resources) to elaborate on these documents.

As such, this choice can contribute to the sharing of know-how about local climate change adaptation with the neighboring municipalities.

It was also possible to verify that strategies that were elaborated in the scope of the ClimAdaPT project present in detail their adaptation options, even ranking these options by priority order. However, the other strategies made by the municipalities that were not participants in this project were not so clear in their adaptation measures, with their descriptions being made very superficially.

### 5.2. Questionnaires to the Coastal Municipalities

Through the results from the questionnaire, it was also possible to conclude that half of the coastal municipalities answered it. The lack of response to this questionnaire might signify underlying problems, namely the number of public administration employees of each municipality. As is understandable, bigger municipalities, or municipalities in certain areas of influence, have more employees and, as such, deploy more human resources to either answer the inquiry or participate in the team that will work on the strategies.

Regarding the answers to the questionnaire, it was possible to expect some of them, due to the previous documentary analysis. These answers, even if expected, were also able to help the authors identify the knowledge of municipalities towards the concepts in the study and if the concepts were included in their respective adaptation strategies. Some answers also allowed the authors to identify inconsistencies, which will be explored further.

Moreover, it was possible to verify that almost half of the municipalities that responded to the inquiry have an adaptation strategy and approximately 40% have their own strategy in development. For municipalities that do not have adaptation strategies, four out of the six municipalities are included in regional or inter-municipal climate change adaptation plans. Several reasons were given by the municipalities for the lack of individual strategies, such as a lack of human resources and lack of funding. As mentioned before, these responses can relate to the number of employees in the local administration of municipalities.

Data collected from PorData show that, as of 2020, there were only 135,125 people working in local administration in Portugal. The Azores Islands have 2878 local administration employees, while the Madeira islands have 3133. Moreover, the Centro region has 29,242 employees, and the Algarve has 9733. This can contribute to the lack of implemented adaptation strategies in municipalities from this NUTS II.

By exploring the data of the municipalities that do not have strategies implemented on an individual scale, it is possible to see the number of employees in local administration in these areas (Table 5).

For the municipalities in the islands, the number of public employees is justifiable. However, for the mainland municipalities, the data shows there could be enough human resources for the development of an adaptation strategy.

Another reason these municipalities give for not having an adaptation strategy is that it is not planned, or that there is no interest in it. This may reveal a lack of interest in the executive regarding climate laws, namely the New Climate Law of 31^st^ of December of 2021 which requires municipalities to have a municipal or regional adaptation strategy implemented.

Results from the municipalities that have adaptation strategies in development allowed us to verify a clear interest from municipalities in developing an adaptation strategy to comply with the New Climate Law and to implement measures that can adapt these municipalities and avoid a higher risk in face of climate change.

As for the municipalities that do have adaptation strategies, most of the documents were approved in 2016 and 2017. This also derives from the ClimAdaPT program, which took place between January 2015 and December 2016. Of the six strategies approved in 2016, four were from municipalities that participated in this program, and from the three strategies approved in 2017, two were also from participating municipalities.

The fact that some municipalities have strategies that are not on a municipal/individual level is concerning, mainly because it means these documents are not as specific as they ought to be since the role of the municipality is not considered with such dept when integrated into an inter-municipal community or in a metropolitan region. As such, it is necessary that these municipalities develop adaptation strategies on a municipal scale, making them more specific and in accordance with local problems and with common solutions led by the different stakeholders that are relevant to the municipality, while simultaneously ensuring active participation throughout the whole process.

The information from the results allows us to conclude that these municipalities have an overall knowledge of the concept of NBS and GI, and adaptation options that fall into the latter concept are included in all strategies. This will ensure that there is less risk of damage in the municipalities, either from a social, environmental, or economic perspective—which can also be interconnected.

Crossing data from the results of the documentary analysis with the results of the questionnaire, regarding NBS and GI, it is possible to verify that two municipalities answered that they have contemplated GI and NBS in their strategies. However, analyzing the strategies from these two municipalities, GI and NBS are not mentioned in any of them and in one of the municipalities, the adaptation options are implicit. It is important to note that these two municipalities did not participate in the ClimAdaPT project and, as such, followed a different structure for their respective documents.

However, the results concerning CBA are not optimistic. A large percentage of municipalities have no knowledge of this concept and did not include it in the adaptation strategy.

The participation of the community is a relevant part of the success of climate change adaptation, and it is acknowledged by the municipalities in their strategies and in the answers to the questionnaire. The members of the community can contribute with different knowledge and that can contribute to better solutions to the problems. Nevertheless, the authors also recognize that the participation of the community has a number of constraints associated: the lack of involvement of people and internal divisions in the community.

Therefore, there is a knowledge gap in the Portuguese coastal municipalities regarding CBA. It must be addressed for the adaptation strategies to succeed and include adaptation measures that come from it.

## 6. Conclusions

This research had the main purpose of addressing climate change adaptation strategies at a local scale, using Portuguese coastal municipalities as a case study. Moreover, this research aimed to understand the inclusion of the local communities in the elaboration of these strategies.

From questionnaires addressed to the Portuguese coastal municipalities, it was possible to understand that most municipalities have climate change adaptation strategies implemented or in development (86.7% of the inquired municipalities). For the municipalities that do not have strategies, the main reason given was the lack of human resources to work on this document. This also connects to the evaluation made by the European Commission in 2018, regarding the Member States’ adaptation preparedness. For Portugal’s fiche, it was assessed that few adaptation plans were moving onto the implementation stage—with this being the weakest part of the process, along with monitoring and evaluating policies [30]. It was also concluded that there is a deep knowledge of nature-based solutions and green infrastructure among the municipalities, with most adaptation strategies having adaptation options and measures that reference these notions.

However, a concerning conclusion was made through the obtained results: most coastal municipalities in Portugal have little knowledge of the concept of community-based adaptation. In fact, because it is such a recent concept, there are still not many scientific articles published on this topic, most of them being studies regarding the indigenous population and their knowledge of nature [62,63,64].

The engagement of the community in local processes is essential for the success of public policies. This is also applied to climate change adaptation since the different local stakeholders have knowledge of different themes, which can contribute to solutions that better serve the community. As such, it is important to invest in the education of the local public administration regarding CBA, to ensure the engagement of people in the decision-making process. In this process, it would be relevant to include stakeholders such as R&D institutions and universities, as well as companies and the community [36]. This can be accomplished through short-term mandatory courses or seminars aimed mainly at employees that work directly with these policies or, ultimately, through law enforcement—as a more serious measure required by the executive power.

One example of the engagement of the local community in climate change adaptation in Portugal is the PLAAC-Arrábida Project. In this project, three municipalities, with the coordination of one energy agency and the collaboration of two universities and their local stakeholders, have successfully developed three local climate change adaptation strategies. Over the course of 15 months, all parties continuously collaborated in three different meetings and five different workshops to discuss which local community members should also be considered for this cooperation, to identify the main areas at risk, to elaborate the specific actions and measures for climate change adaptation and, finally, to choose the most relevant ones for their specific territory.

This project, and the results obtained throughout this research, exhibit the importance of community participation for the success of climate change adaptation at a local level, and should, therefore, be a driver for all coastal communities worldwide. Thus, it is crucial that more CBA studies are undertaken to consolidate a framework for its use in different places and communities globally.

## Figures and Tables

**Figure 1 ijerph-19-16687-f001:**
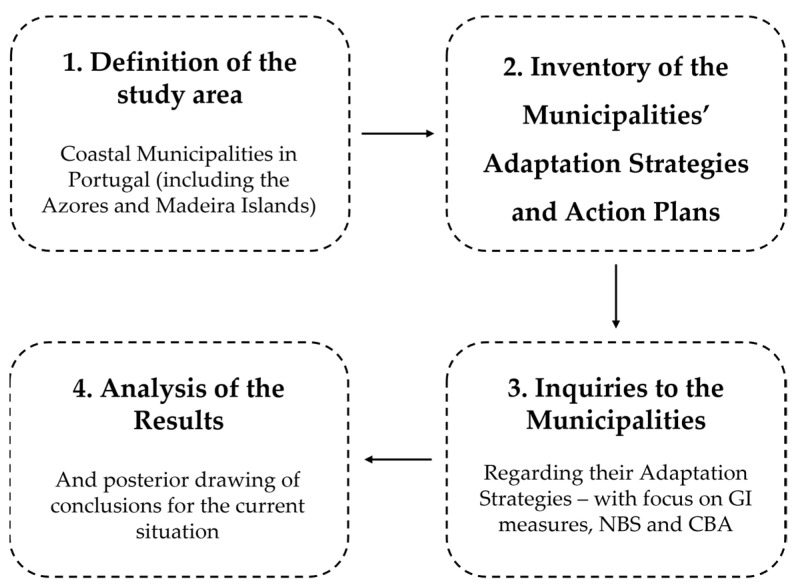
Methodological Framework.

**Figure 2 ijerph-19-16687-f002:**
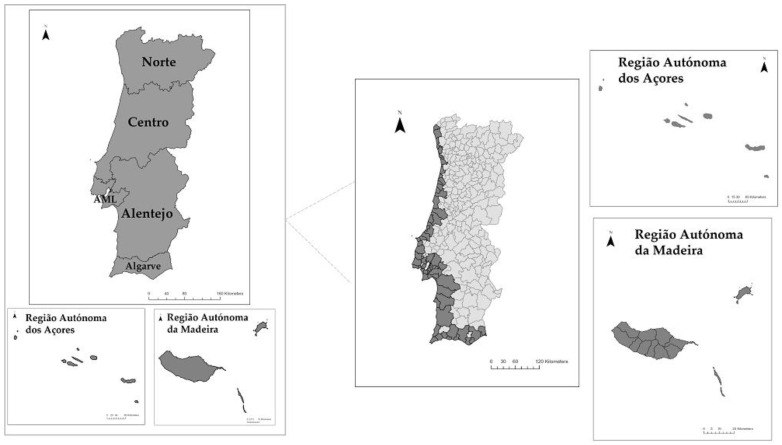
NUTS II and study area (coastal municipalities), where AML is Lisbon Metropolitan Area (Data from: CAOP 2021).

**Figure 3 ijerph-19-16687-f003:**
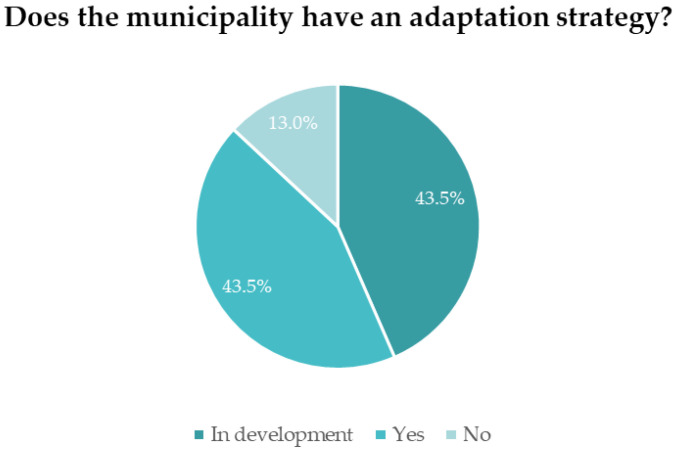
Percentage of surveyed municipalities that have climate change adaptation strategies.

**Figure 4 ijerph-19-16687-f004:**
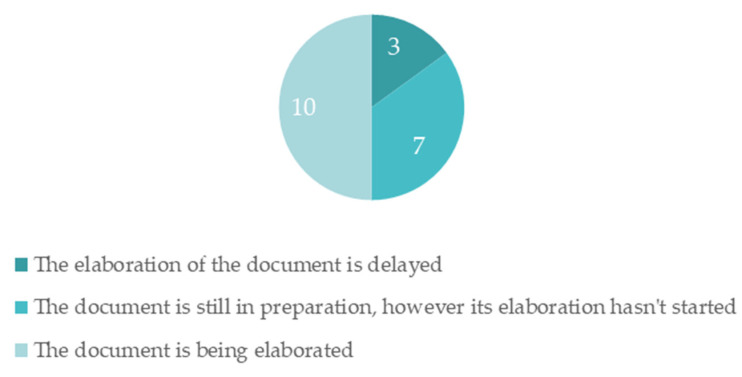
Municipalities with strategies in development.

**Figure 5 ijerph-19-16687-f005:**
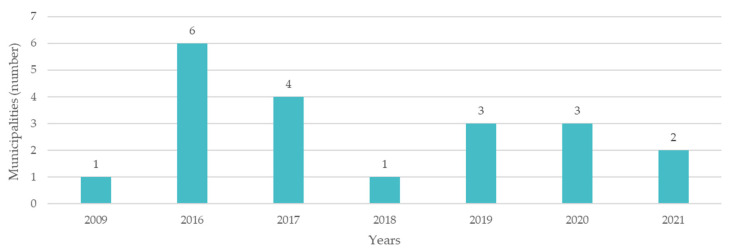
Climate change adaptation strategies approved by year.

**Figure 6 ijerph-19-16687-f006:**
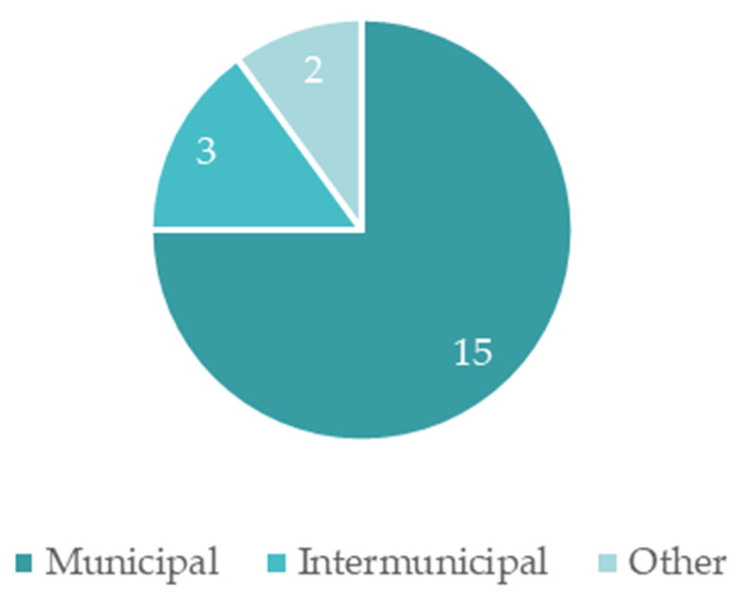
Scale of action of the adaptation strategies.

**Figure 7 ijerph-19-16687-f007:**
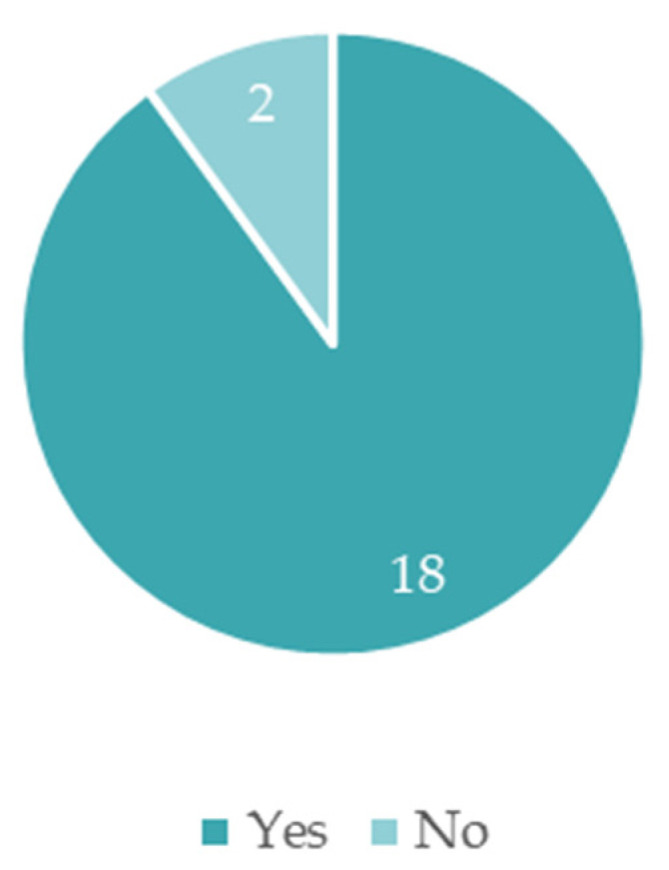
Nature-based solutions in adaptation strategies.

**Figure 8 ijerph-19-16687-f008:**
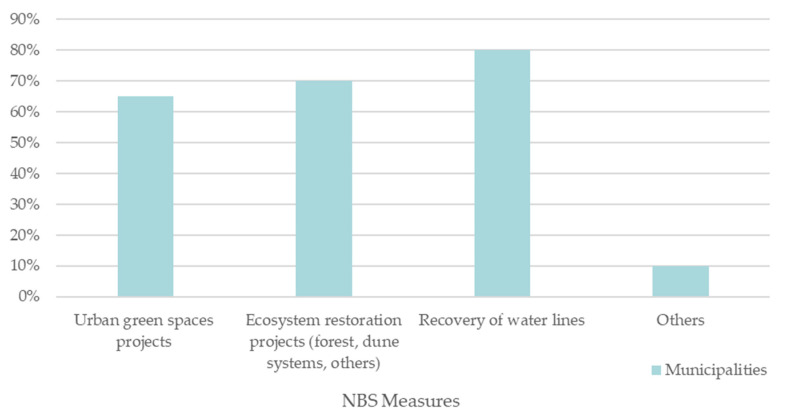
Main NBS measures in the adaptation strategies.

**Figure 9 ijerph-19-16687-f009:**
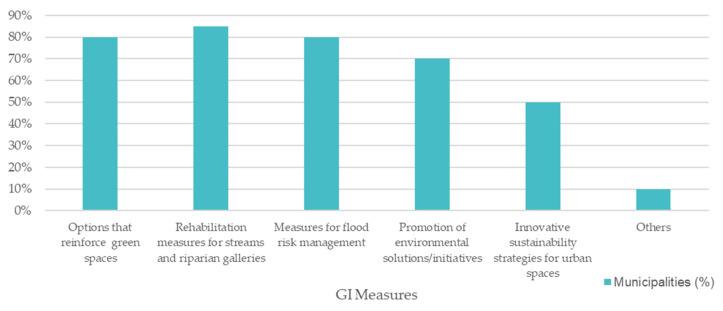
GI measures mentioned in the adaptation strategies.

**Figure 10 ijerph-19-16687-f010:**
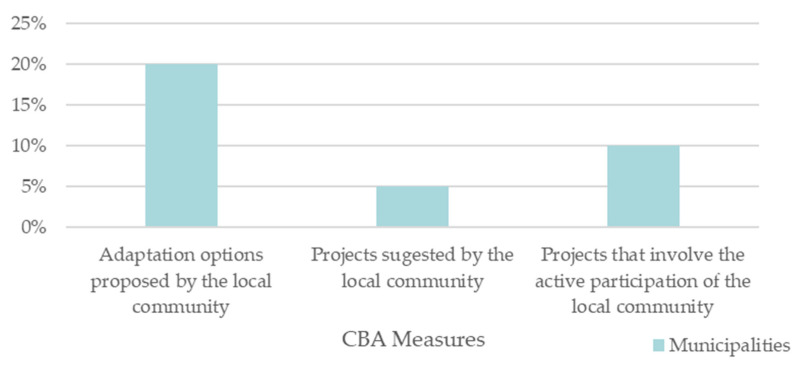
CBA measures mentioned in the adaptation strategies.

**Table 1 ijerph-19-16687-t001:** Municipalities with individual adaptation strategies, by NUTS II.

NUTS II	Municipalities
Norte	Viana do Castelo
	Esposende
	Vila do Conde
	Porto
	Espinho
Centro	Aveiro
	Ílhavo
	Leiria
	Torres Vedras
Lisbon Metropolitan Area	Mafra
	Barreiro
	Cascais
Alentejo	Benavente
	Odemira
Algarve	Loulé
	Faro
Região Autónoma da Madeira	Funchal
Região Autónoma da Açores	Vila Franca do Campo

**Table 2 ijerph-19-16687-t002:** Municipalities that do not have strategies and their corresponding NUTS II.

NUTS II	Municipalities
Centro	Marinha Grande
	Vagos
Algarve	Olhão
Região Autónoma dos Açores	Horta
	Santa Cruz da Graciosa
Região Autónoma da Madeira	Câmara de Lobos

**Table 3 ijerph-19-16687-t003:** Municipalities that have strategies in development and their corresponding NUTS II.

NUTS II	Municipalities
Região Autónoma dos Açores	Praia da Vitória
	Ponta Delgada
	Angra do Heroísmo
	Povoação
	Velas
	Madalena
	Santa Cruz das Flores
Alentejo	Santiago do Cacém
Algarve	Tavira
	Albufeira
	Castro Marim
	Silves
Centro	Ovar
	Caldas da Rainha
	Nazaré
Lisbon Metropolitan Area	Oeiras
	Seixal
	Sesimbra
Região Autónoma da Madeira	Calheta
	Ribeira Brava

**Table 4 ijerph-19-16687-t004:** Municipalities that have strategies and their corresponding NUTS II.

NUTS II	Municipalities
Norte	Esposende
	Viana do Castelo
	Vila do Conde
	Matosinhos
	Vila Nova de Gaia
	Porto
Centro	Ílhavo
	Figueira da Foz
	Leiria
	Óbidos
	Torres Vedras
Área Metropolitana de Lisboa	Almada
	Cascais
	Lisboa
	Loures
	Setúbal
	Sintra
Algarve	Loulé
	Faro
Região Autónoma dos Açores	Ribeira Grande

**Table 5 ijerph-19-16687-t005:** Number of local public administration employees (source: PorDATA).

Municipality	Number of Local Public Administration Employees
Marinha Grande	266
Vagos	247
Olhão	593
Horta	54
Santa Cruz da Graciosa	166
Câmara de Lobos	248

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
