# Peer review of "Climate Change Adaptation Strategies at a Local Scale: The Portuguese Case Study"

_ijerph, 2022, doi:10.3390/ijerph192416687_

Round 1

Reviewer 1 Report

The authors use document analysis and questionnaire to explored the climate change adaptation strategies in the coastal areas in the Portuguese. Although the research method is simple, the content is significant. There have some problems should be revised as follow.

1.There are some existing concept of climate change adaptation. How to define climate change adaptation in this manuscript by authors.

2.In 2.3 section, how to utilize the natured-based, green infrastructure and CBA to interfere climate change? What's the relationship of climate and them? What is the compose of natured-based, green infrastructure and CBA? The authors should illustrate them detailed.

3.What’s the difference of GI and natured-based?

4.The authors have the map of study area. It is necessary to analyze the spatial of climate change adaptation strategies. It can make the work full of significant. This manuscript only statistic the number of climate change adaptation strategies. It is too simple.

5. The authors illustrated eight actions in line 72. It should be illustrated further. Because some specific is difficult to understand, such as one-stop shop.

6.The presentation of English is a little poor. Some sentences are lack of logic. Such as line 29-31, line 61-65.

7.What is the meaning of local and urban adaptation in line 88?

8.The quality of the figures should be improved. On the one hand, the figures should be colored. On the other hand, there have two captions of figures.

9.In line 198, the authors think urban heat island effect by cooling urban spaces, especially in population dense locations. I suppose there should have some relevant references to support this opinion. I recommend the references as follow:

Relationships among local-scale urban morphology, urban ventilation, urban heat island and outdoor thermal comfort under sea breeze influence. Sustainable Cities and Society,2020,60:102289. doi: https://doi.org/10.1016/j.scs.2020.102289.

Spatiotemporal relationship characteristic of climate comfort of urban human settlement environment and population density in China. Front. Ecol. Evol. 2022, 10:953725.
doi: 10.3389/fevo.2022.953725

Understanding land surface temperature impact factors based on local climate zones. Sustainable Cities and Society,2021,69:102818. doi: https://doi.org/10.1016/j.scs.2021.102818

Author Response

The replies to the comments and suggestions of Reviewer 1 can be found in the attached .PDF file.

Reviewer 2 Report

The article is structured and very clear in its wording, however, it has unfortunately a couple of major weaknesses in content.

The introduction is very short. The state of the art is treated very superficially and does not provide enough background information; the relevant references must definitely be added if the paper is to be accepted for publication.

The authors criticise that there is no consensual definition of climate change adaptation, but do not present the concepts or possibly divergent concepts in more detail. This needs to be better elaborated. Also, in my opinion, it is not sufficient to prove with a single scientific source that climate change adaptation is more successful when applied at the local scale. This argument or presupposition also needs to be elaborated and requires further scientific evidence.

The authors also do not sufficiently explain why they focus on Nature-based  Solutions (NbS), Green Infrastructure (GI) and community-based adaptation (CBS). These concepts are also treated only very superficially and also very uncritically. While there’s a lack of distinction between NbS and GI, challenges in CBA practice are not reflected critically (e.g. misperceptions of the notion of ‘community’, perils of participation, etc.).

The methodology is kept quite straightforward and also the questions to the municipalities are unfortunately quite simple - accordingly, the results are predictable and expectable. The study does not deliver any surprising results and the text is mainly limited to whether or not municipalities have an adaptation strategy. 

The presentation of the results (analysis?) as well as the discussion and conclusions on NbS, GI and CBA are very short in content and remain superficial (line 611-622).

Author Response

(The authors gave the same response as above.)

Round 2

Reviewer 1 Report

All the problems have been addressed. There only have some minor problems.

1. What's the meaning of MR2 in line 29.

2.In line 266, Al should be revised to al.

3.There have two title in each of figures. It should be revised.

Author Response

We are grateful to the reviewer for the dedication given in revising the paper for a second time. 

Regarding your questions, below you can find the answers to your questions:

  1. We cannot find "MR2" on the line mentioned. However, it was probably a commentary made by one of the authors. Nonetheless, it has been deleted and the final manuscript does not include this.
  2. Thank you! It has already been changed.
  3. The titles of the graphs have been removed, since they are already written in the caption of the figure. Thank you for the commentary.

Reviewer 2 Report

Thank you for the revision of the manuscript. It is now much improved.
I can understand very well from my own experience that data collection must have been very difficult in times of the Pandemic. However, my concerns about the straightforwardness of the interview questions and the resulting "predictability" of some of the answers have not yet been fully resolved.

Author Response

We are grateful to the reviewer for the dedication given in revising the paper for a second time. 

Regarding the answers to the questionnaire, it was possible to expect some of them, due to the documental analysis made in the scope of this research. These answers, even if expected, were also able to help the authors identify the knowledge of municipalities towards the concepts in study and if those concepts were included in their respective adaptation strategies. Some answers also allowed the authors to identify inconsistencies, which will be explored ahead in this discussion.